# Impact of QRS misclassifications on heart-rate-variability parameters (results from the CARLA cohort study)

Frank Sauerbier[1], Johannes Haerting[1], Daniel Sedding[2], Rafael Mikolajczyk[1], Karl Werdan[2], Sebastian Nuding[2], Karin H. Greiser[3], Cees A. Swenne[4], Jan A. Kors[5], Alexander Kluttig[1]*

**1** Institute of Medical Epidemiology, Biometrics, and Informatics, Interdisciplinary Center for Health Sciences, Medical Faculty of the Martin-Luther-University Halle-Wittenberg, Halle (Saale), Germany, **2** Department of Internal Medicine III, University Hospital, Martin-Luther-University Halle-Wittenberg, Halle (Saale), Germany, **3** Division of Cancer Epidemiology, German Cancer Research Center, Heidelberg, Germany, **4** Cardiology Department, Leiden University Medical Center, Leiden, The Netherlands, **5** Department of Medical Informatics, Erasmus MC, University Medical Center Rotterdam, Rotterdam, The Netherlands

* alexander.kluttig@uk-halle.de

**Data Availability Statement:** Due to ethical constraints individual data cannot be made publicly available. The data that support the findings of this study are available on request from the Institute of

## Abstract

### Background

Heart rate variability (HRV), an important marker of autonomic nervous system activity, is usually determined from electrocardiogram (ECG) recordings corrected for extrasystoles and artifacts. Especially in large population-based studies, computer-based algorithms are used to determine RR intervals. The Modular ECG Analysis System MEANS is a widely used tool, especially in large studies. The aim of this study was therefore to evaluate MEANS for its ability to detect non-sinus ECG beats and artifacts and to compare HRV parameters in relation to ECG processing. Additionally, we analyzed how ECG processing affects the statistical association of HRV with cardiovascular disease (CVD) risk factors.

### Methods

20-min ECGs from 1,674 subjects of the population-based CARLA study were available for HRV analysis. All ECGs were processed with the ECG computer program MEANS. A reference standard was established by experienced clinicians who visually inspected the MEANS-processed ECGs and reclassified beats if necessary. HRV parameters were calculated for 5-minute segments selected from the original 20-minute ECG. The effects of misclassified typified normal beats on i) HRV calculation and ii) the associations of CVD risk factors (sex, age, diabetes, myocardial infarction) with HRV were modeled using linear regression.

### Results

Compared to the reference standard, MEANS correctly classified 99% of all beats. The averaged sensitivity of MEANS across all ECGs to detect non-sinus beats was 76% [95% CI: 74.1;78.5], but for supraventricular extrasystoles detection sensitivity dropped to 38%

Medical Epidemiology, Biometrics and Informatics (IMEBI)(Email: imebi@uk-halle.de) or using a form on the CARLA website https://webszh.uk-halle.de/carla-studie/. The data are not publicly available due to restrictions: the data contains personally identifiable data on human research participants such as specific dates (birth dates, death dates, examination dates, etc.). Those interested can access the data in the same manner as the authors. The authors had no special access privileges to the data. MEANS algorithm has been licensed by biomedical companies and is not freely available.

**Funding:** The CARLA Study was funded by a grant from the Deutsche Forschungsgemeinschaft (DFG) as part of the Collaborative Research Centre 598 'Heart failure in the elderly – cellular mechanisms and therapy', by an additional single funding grant from the DFG, by three grants of the Wilhelm-Roux-Program of the Martin-Luther-University of Halle Wittenberg (FKZ 14/41, 16/19 and 28/21), by the Federal Employment Office, by the Ministry of Education and Cultural Affairs of Saxony-Anhalt (MK-CARLA-MLU-2011). The project "Biosalsa" (project ID: ZS/2019/07/99752) belongs to the Research Association „Autonomy in old Age" (AiA) funded by the European Union (ERDF European Regional Development Fund) and the State of Saxony-Anhalt, Germany.

**Competing interests:** The authors have declared that no competing interests exist.

[95% CI: 36.8;38.5]. Time-domain parameters were less affected by false sinus beats than frequency parameters. Compared to the reference standard, MEANS resulted in a higher SDNN on average (mean absolute difference 1.4ms [95% CI: 1.0;1.7], relative 4.9%). Other HRV parameters were also overestimated as well (between 6.5 and 29%). The effect estimates for the association of CVD risk factors with HRV did not differ between the editing methods.

## Conclusion

We have shown that the use of the automated MEANS algorithm may lead to an overestimation of HRV due to the misclassification of non-sinus beats, especially in frequency domain parameters. However, in population-based studies, this has no effect on the observed associations of HRV with risk factors, and therefore an automated ECG analyzing algorithm as MEANS can be recommended here for the determination of HRV parameters.

## 1 Introduction

The rapid technological development in the field of wearables makes it increasingly easy to collect health data such as heart rhythm and even the electrocardiogram (ECG) [1] from a large number of people in the general population. Heart rate variability (HRV) is considered an important measure of autonomic nervous system activity. A large number of studies have demonstrated the prognostic significance of various HRV parameters, especially for total and cardiovascular-specific mortality [2]. Further, they are related to risk factors for cardiovascular and metabolic diseases [3] and can be influenced by lifestyle interventions [4]. HRV can also play an important role in the long-term monitoring of Covid patients [5, 6]. To calculate HRV parameters, RR-interval data of a pure sinus rhythm are necessary. However, ECGs often include ectopic beats or artifacts that should be recognized before HRV analysis. Usually, in a clinical setting detection of non-sinus beats is done by experienced cardiologists or at least trained staff. However, this approach is time-consuming and costly and therefore difficult to apply in large population-based studies. Therefore, a variety of algorithms were developed to classify heartbeats automatically. An overview of many algorithms and their methods can be found in Luz et al., 2016 [7]. For evaluation and reporting of the performance of those algorithms in detecting the correct beat types, recommendations were given more than 30 years ago by ANSI/AAMI (Association for the Advancement of Medical Instrumentation in cooperation with the American National Standards Institute), updated in 2013 [8].

Some literature deals with the influence of non-corrected ectopic beats on HRV calculation. This was investigated in several studies [9, 10]. However, to date, no study has evaluated the impact of non-detected non-sinus beats on the association of HRV with well-known CVD risk factors.

Therefore, our aim was i) to evaluate the performance of a widely used algorithm in the classification of heartbeats in a general population, ii) to quantify the effect of undetected non-sinus ECG beats on the calculation of HRV parameters, and iii) to analyze how these misclassifications affect the statistical association of HRV with CVD risk factors.

The "Modular ECG Analysis System" (MEANS) is a validated [11] and widely used automatic algorithm in population-based studies, e.g., the Rotterdam study [12], the HELIUS study [13], the SHIP study [14], and the NAKO study [15]. The development of the program began

several decades ago. It has a completely modular structure. Individual modules could therefore be entirely redeveloped without disrupting the overall framework. In a testing of nine different programs, MEANS showed excellent results [16]. However, the dependence of the calculated HRV on such an ECG analysis system and, in particular, the dependence of association studies involving HRV, have not yet been investigated. We, therefore, decided to perform the analysis using the MEANS algorithm as an important example for heartbeat classification algorithms.

## 2 Materials and methods

The analyses are based on data from the baseline examination of the prospective, population-based CARLA study (*CAR*diovascular disease, *Li*ving and *A*geing in Halle). The original aim of the CARLA study was to analyze the causes of increased cardiovascular morbidity and mortality in the region and to investigate the importance of HRV as a marker of autonomic function and predictor of cardiovascular events. Details of the study have been described elsewhere [17–19]. In brief, the CARLA study is a prospective cohort study of a representative sample of the elderly inhabitants of the city of Halle (Saale). A random sample of 5.000 people aged 45 to 80 years at the time of the sampling (July 2002) was drawn from the population registry of the city of Halle. The recruitment of study subjects has been done by inviting consecutive waves of random sub-samples of the original population sample. Accordingly, not all persons originally drawn from the population registry had to be invited in order to achieve a representative target sample of 1750 subjects of the Halle population aged 45–80 years. Of the 3437 subjects invited to participate in the study, 1779 participants aged 45–83 years at baseline were recruited, of which 812 (46%) were women and 967 (54%) men, resulting in a final response proportion of 64.1% after exclusion of persons who deceased prior to the invitation, moved away or were unable to participate due to illness. The recruitment of study subjects and the baseline examination began in December 2002 and ended in January 2006. All participants gave their written informed consent.

### 2.1 ECG recording and processing

Fig 1 shows the basic steps of data collection, processing, and analysis. First 20-min 12-lead resting ECG (CardioControl Working Station, Welch Allyn, Delft, the Netherlands) was recorded in the supine position after a resting period of at least 10 minutes. Throughout the ECG recording (performed with a sampling frequency of 600Hz), subjects were asked to breathe at 15 respirations per minute (= 0.25Hz, guided by a visual metronome) to standardize the effects of respiratory rate on HRV.

All ECGs were processed by MEANS to obtain the location and type of the QRS complexes [11]. MEANS is a well-validated and widely used automated ECG processing and interpretation system [16, 20, 21]. A detailed description and comparison of the MEANS algorithm with other algorithms is outside the scope of our article. However, the MEANS algorithm has extensively been described and evaluated before [22, 23], showing its validity in correct classification of QRS complexes. For example, on the CSE multi-lead library containing 250 ECGs, all 2,081 QRS complexes were correctly detected, and 99.6% of the complexes were correctly typified [22].

As already mentioned, MEANS has a completely modular structure. Signal evaluation is mainly carried out serially by the modules. In addition to the input and output modules, 4 further module groups are responsible for the following tasks [11]:

- detection of QRS complexes and disturbances as well as the P waves

- typification according to waveform for the QRS or ST-T complexes

- determination of the dominant complexes for contour analysis

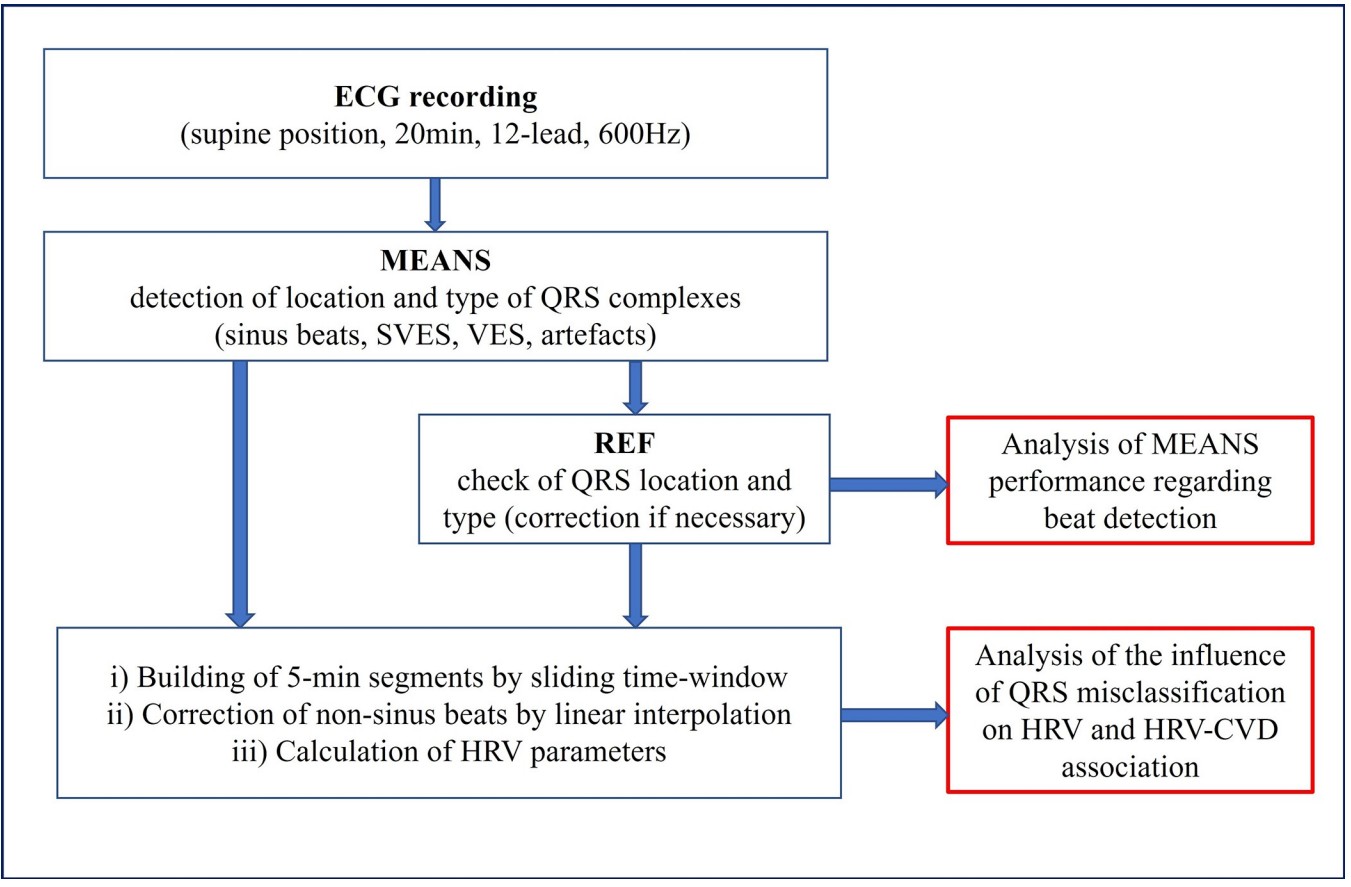

**Fig 1. Flow chart of the basic sequence of the individual processing steps for determining HRV values for MEANS and REF.**

- segmentation of P and T waves and QRS complexes (recognition of start and end points)

- classification (rhythm, contour) of the ECG.

Afterward, all MEANS-processed ECGs were visually controlled for the correctness of determination of QRS location and type by two trained medical students supervised by an experienced cardiologist (in the following called "reference" = REF).

We differentiate three types of QRS complexes, normal sinus beats, supraventricular extrasystoles (SVES), and ventricular extrasystoles (VES). MEANS considers any abnormally typified QRS complex as ventricular, while the distinction between normal and atrial complexes is based on the timing and morphology of the preceding P waves. Artifacts were detected and classified by MEANS QRS detector, resulting from sudden baseline shifts or spikes. Finally, we group SVES, VES, and artifacts as non-sinus beats.

For each study participant, both the 20-min tachograms of the MEANS and the REF-controlled ECGs were then separately used for HRV calculation according to the following procedure. First, sixteen 5-min segments of each of the 20-min tachograms were produced by a sliding 5-min window moving over the entire tachogram in 1-minute steps. If in a 5-min MEANS or REF segment the proportion of non-sinus beats or artifacts was greater than 10%, both the MEANS and REF segments were removed. If all segments of an ECG contained more than 10% non-sinus beats, the ECG was excluded from this analysis. Additionally, the remaining segments were checked for stationarity of the RR intervals according to the reverse

arrangement test at the 5% level to means and variances and removed when the stationarity z-score cut-off exceeded 1.96 [24]. Removal of the segments with non-stationary data did not lead to the exclusion of any of the study participants due to a lack of data. From the remaining segments, the segment of the MEANS-processed ECG with the lowest percentage of non-sinus beats and the corresponding REF segment was selected for HRV analysis. If several choices were possible, the earliest segment occurring in time was chosen.

HRV parameters were calculated for each of the MEANS-analyzed and REF-edited 5-min segments, independent of the selection procedure described above. In each of these segments, first, the non-sinus beats were replaced by interpolated sinus beats, inserted at 50% of the interval between the sinus beats preceding and following the non-sinus beats, thus creating a pseudo-sinus-rhythm interval series. Then, like in Bootsma et al., 2003 [25], the HRV spectrum was calculated from this non-equispaced interval data series, without resampling, as follows:

- normalization of the interval series by dividing the intervals by the average interval

- subtraction of the linear trend in the normalized interval series (creates a series of interval fluctuations around zero)

- cosine tapering of the initial and terminal 10% of this data series (creates a fade-in and fade-out effect, thus preventing an unwanted invasion of the spectrum by data-on and data-off transients)

- adding zeroes to the normalized, detrended, and tapered interval fluctuation series till the number of data in that series equals a power of 2 (a requirement for the FFT algorithm)

- computation of the FFT spectrum of this data series

- correction of the frequency spectrum for the energy loss as a consequence of the cosine tapering and zero padding, by multiplying it by a factor larger than 1 in order to make the total power of the frequency spectrum identical to the variance of the original time series (thus satisfying Parseval's theorem).

Standard time-domain (SDNN in ms), pNN50 (percentage of successive RR intervals that differ by more than 50ms), RMSSD (root mean square of successive RR interval differences in ms) and frequency-domain parameters (LF (absolute power of the low-frequency band (0.04–0.15Hz) in $ms^2$), HF (absolute power of the high-frequency band (0.15–0.4Hz) in $ms^2$), LF/HF (ratio of LF-to-HF power)) were computed according to the method used in previous studies [25–28] and in compliance with the current guidelines for analysis of HRV [29].

## 2.2 Statistical methods

Performance analyses of MEANS regarding the correct detection of non-sinus beats were done using the entire 20-min ECGs. The sensitivity and specificity of the non-sinus beat detection performance of MEANS were determined according to the AAMI guidelines [8]. Two types of aggregated statistics are given: in the "gross statistics", all beats are weighted equally, regardless of whether they originate from one or the other ECG. The dependency of beats from one ECG is therefore not taken into account. This is in contrast to the average statistics, in which the dependency of beats from an ECG is taken into account and beat statistics are calculated individually for each ECG and then averaged.

Bland-Altman plots were used to visualize the differences between the HRV measures resulting from the different editing methods. As the differences did not seem to be normally distributed we took the logarithmically transformed data [30].

To evaluate the influence of misclassified beats on the HRV parameters, we used linear regression models with the percentage of false sinus beats as the independent variable and the relative HRV difference between MEANS and REF as the dependent variable (here the non-logarithmized data were used):

$$(HRV_{MEANS} - HRV_{REF})/HRV_{REF}.$$

The association between selected CVD risk factors and the logarithmically transformed HRV values was assessed by linear regression models for both editing methods. Analyzed CVD risk factors were sex, age, previous myocardial infarction (MI), and prevalent type 2 diabetes mellitus. In the case of age, a quadratic term was additionally taken into account to reflect the non-linear relationship between age and HRV in our study [18]. Beta values and 95% confidence intervals (CI) for the correlation of the logarithmized HRV values of the respective editing method were obtained directly from the data, whereas the betas and the CI of the difference were obtained by bootstrapping beforehand, utilizing the non-parametric bootstrapping method with 10,000 resamples and a fixed seed. Nonparametric bootstrapping allows reliable estimation of statistical parameters, especially CI, without model assumptions. It is a kind of resampling procedure. Samples of the same size as the original sample are taken from the distribution of values of interest, with replacement. In each of these new samples, the statistics of interest are calculated, and the CI sought is estimated from the distribution of these results. The accuracy of this estimate depends on the quality of the initial sample. We use the percentile approach to calculate the CI, which is a very simple method but gives good results for approximately symmetric distributions [31].

Of the 1,779 participants of the CARLA study, 1,674 had an evaluable 20-min ECG (non-sinus rhythm or pacemaker subjects were excluded, n = 105), which was analyzed by both MEANS and REF. Further 33 ECGs were excluded because of more than 10% non-sinus beats in all 5-min segments, resulting in 1,641 participants remaining for evaluation of the 5-minute HRV.

All statistical analyses were performed using SAS 9.4 University Edition [32].

## 3 Results

The baseline characteristics of the cohort are shown in Table 1. Overall, our study group showed a high burden of CVD risk factors and diseases.

Based on a total of more than 2.2 million beats of the 1,674 20-min ECGs, we found an overall agreement of MEANS and REF in the beat classification of 99.05% [95% CI: 99.04; 99.06]. The overall sensitivity of MEANS in detecting non-sinus beats was 65.5% [95% CI: 64.9; 66.1] in gross statistics. Table 2 shows the performance of MEANS subdivided according to different beat types. In detail, the MEANS algorithm failed mainly in correctly classifying SVES (supraventricular extrasystoles) and artifacts. About one-third (7.284 of 21.580 beats) of erroneously determined beat types were SVES, typified by MEANS as sinus beats, and more than half (12.135 of 21.580 beats) were sinus beats typified as an artifact. However, 90% of all beats incorrectly classified as sinus beats were found in only 6% of all ECGs. The resulting statistics are shown in Table 3, separated into gross and average statistics. The values are consistently higher for the averaged statistic than for the gross statistic, especially those for the sensitivity. In particular, the sensitivity for SVES is almost doubled in the averaged approach, which can be explained by the low proportion of ECGs with frequent ectopic beats and artifacts.

Fig 2 shows the Bland-Altman plot of HRV derived via MEANS versus REF exemplarily for SDNN. The corresponding plots for RMSSD, pnn50, LF, and HF (not shown) manifest a

**Table 1. Baseline characteristics of subjects included in ECG analyses (1641 subjects).**

| | Women | | | Men | | |
|---|---|---|---|---|---|---|
| | N | $\bar{x}$ or % | 95% CI | N | $\bar{x}$ or % | 95% CI |
| Age (yrs) | 768 | 63.3 | 62.6;64.0 | 873 | 64.1 | 63.4;64.7 |
| Weight (kg) | 768 | 73.5 | 72.5;74.6 | 873 | 84.2 | 83.3;85.1 |
| *Drug use*: | | | | | | |
| Betablockers (%) | 261 | 34.0 | 30.6;37.3 | 264 | 30.2 | 27.2;33.3 |
| Antiarrhythmics (%) | 5 | 0.6 | 0.1;1.2 | 4 | 0.5 | 0.0;0.9 |
| ACE-inhibitors (%) | 235 | 30.6 | 27.3;33.9 | 291 | 33.3 | 30.2;36.5 |
| Diuretics (%) | 69 | 9.0 | 7.0;11.0 | 76 | 8.7 | 6.8;10.6 |
| Ca-channelblockers (%) | 113 | 14.7 | 12.2;17.2 | 124 | 14.2 | 11.9;16.5 |
| *Disease prevalence*: | | | | | | |
| Myocardial infarction (%) | 15 | 2.0 | 1.0;2.9 | 72 | 8.2 | 6.4;10.1 |
| Stroke (%) | 21 | 2.7 | 1.6;3.9 | 34 | 3.9 | 2.6;5.2 |
| Cardiovascular disease (CVD)[a](%) | 40 | 5.2 | 3.6;6.8 | 126 | 14.4 | 12.1;16.8 |
| Hypertension[b] (%) | 548 | 71.4 | 68.2;74.6 | 663 | 76.0 | 73.1;78.8 |
| Diabetes mellitus[c] (%) | 109 | 14.2 | 11.7;16.7 | 128 | 14.7 | 12.3;17.0 |

[a]CVD: including prevalent myocardial infarction, coronary artery bypass graft (CABG), percutaneous transluminal coronary angioplasty (PTCA), stroke, carotid surgery

[b]Hypertension defined as SBP $\geq$ 140 and/or DBP $\geq$ 90 mmHg, and/or use of antihypertensive medication by ATC code

[c]Diabetes defined as self-reported physician-diagnosed diabetes mellitus and/or use of anti-diabetic medication by ATC code

similar pattern. In contrast, the pattern for the HRV parameter LF/HF ratio differs (Fig 3). Overall, MEANS resulted in higher HRV values compared to the reference, except for LF/HF. A vast majority of data points lying above the zero line (green diamonds) represent ECGs with prevailing SVES false detections. Two ECGs exhibit a high proportion of non-sinus beat detections that were in reality sinus beats (black circle outlier points).

The impact of undetected non-sinus beats on the calculated HRV values is demonstrated in Figs 4 and 5 for time- and frequency domain parameters (note the different scaling of the y-axis). About 90% of all data points are densely located around zero because of the high percentage of ECGs with none or sparse non-sinus beats. The slopes for the regression line related to time and frequency parameters are given in Table 4. On average, a proportion of only 7% of

**Table 2. Cross table for beat type classification by MEANS and reference standard (SVES, VES: Supraventricular, ventricular extrasystole).**

| | | beat-by-beat performance ("gross statistics") | | | | |
|---|---|---|---|---|---|---|
| | | *REF (visual controlled beat typification)* | | | | |
| **MEANS** | | **Artifact** | **SVES** | **Sinus** | **VES** | **Total** |
| **Artifact** | N | 586 | 68 | 12,135 | 64 | 12,853 |
| | % | 40.61 | 0.56 | 0.54 | 0.62 | 0.57 |
| **SVES** | N | 0 | 4,591 | 10 | 327 | 4,928 |
| | % | 0.00 | 37.61 | 0.00 | 3.18 | 0.22 |
| **Sinus** | N | 668 | 7,284 | 2,220,020 | 307 | 2,228,279 |
| | % | 46.29 | 59.67 | 99.44 | 2.98 | 98.75 |
| **VES** | N | 189 | 265 | 263 | 9,597 | 10,314 |
| | % | 13.10 | 2.17 | 0.01 | 93.22 | 0.46 |
| **Total** | N | 1,443 | 12,208 | 2,232,428 | 10,295 | 2,256,374 |
| | % | 100.00 | 100.00 | 100.00 | 100.00 | 100.00 |

**Table 3. Performance statistics in % for MEANS in beat type detection.**

| Parameter | Non-sinus | 95% CI | SVES | 95% CI | VES | 95% CI | Artifact | 95% CI |
|---|---|---|---|---|---|---|---|---|
| *Gross statistics (calculated without considering individual ECGs)* | | | | | | | | |
| Sensitivity | 65.51 | 64.90–66.11 | 37.61 | 36.75–38.47 | 93.22 | 92.72–93.70 | 40.61 | 38.06–43.20 |
| Specificity | 99.44 | 99.43–99.45 | 99.98 | 99.98–99.99 | 99.97 | 99.97–99.97 | 99.46 | 99.45–99.47 |
| *Average statistics (calculated first within each ECG and then averaged)* | | | | | | | | |
| Sensitivity | 76.29 | 74.06–78.51 | 61.48 | 58.37–64.58 | 97.50 | 96.34–98.66 | 56.24 | 49.33–63.14 |
| Specificity | 99.48 | 99.30–99.66 | 99.98 | 99.95–100 | 99.97 | 99.94–99.99 | 99.50 | 99.32–99.67 |

false detections (corresponding to a beta of about 0.14) produced a doubling of SDNN, for instance. It is remarkable that mostly the time-domain parameters were less susceptible to false sinus beats than the frequency-domain parameters.

The results of the analyses of the association of HRV with CVD risk factors are shown in Table 5. Beta estimators represent the strength of the association between HRV and CVD risk

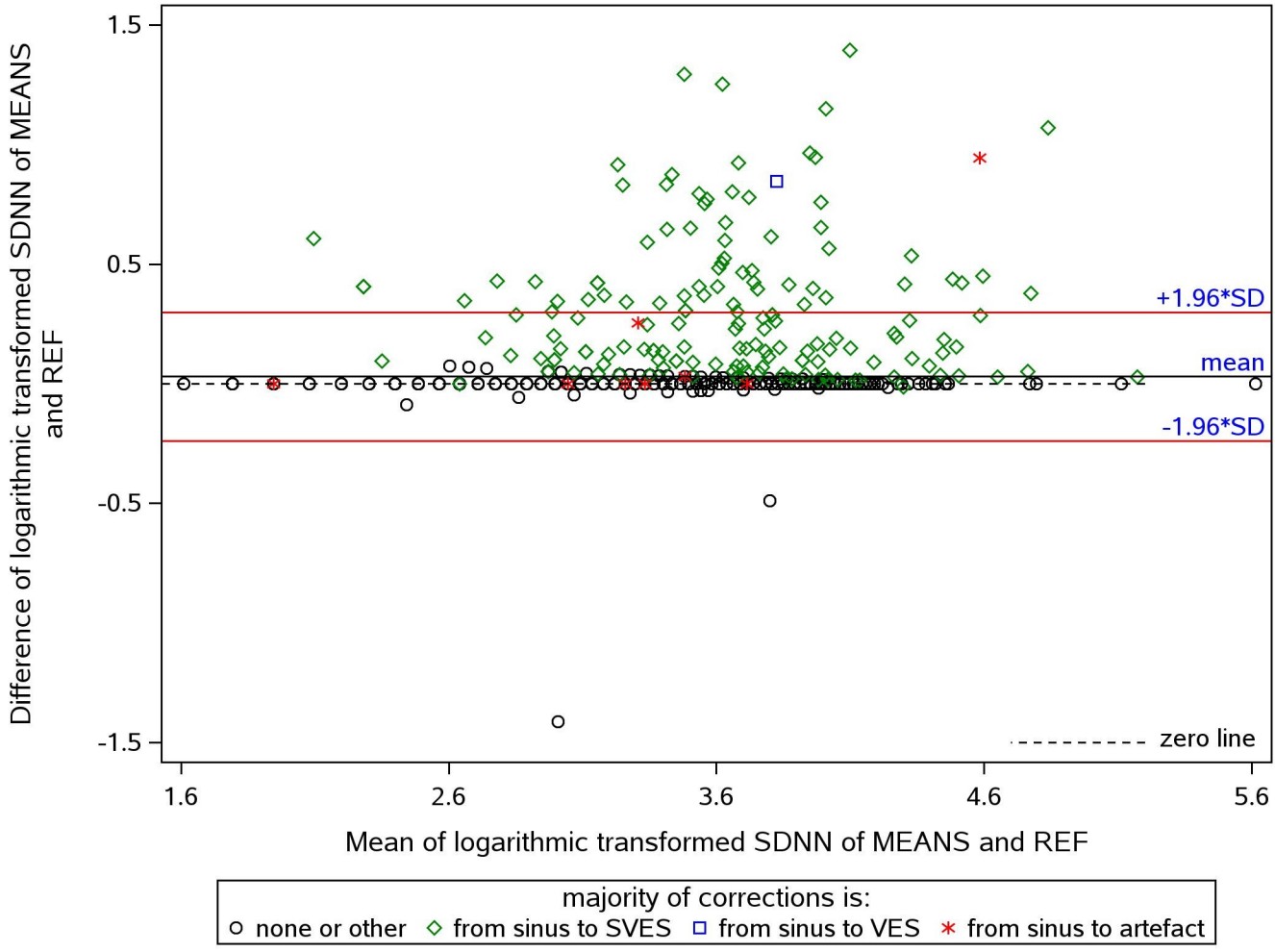

**Fig 2. Bland-Altman plot of the differences (Y-axis) and means (X-axis) of the logarithmic transformed values of SDNN (measured in ms) by MEANS-processed ECG versus reference standard.** Symbols correspond to the majority of beat-type corrections by REF in this 5-min segment. A vast majority of data points lying above the zero line (green diamonds) represent ECGs with prevailing SVES false detections; two ECGs exhibit a high proportion of non-sinus beat detections that actually were sinus beats (black circle outlier points).

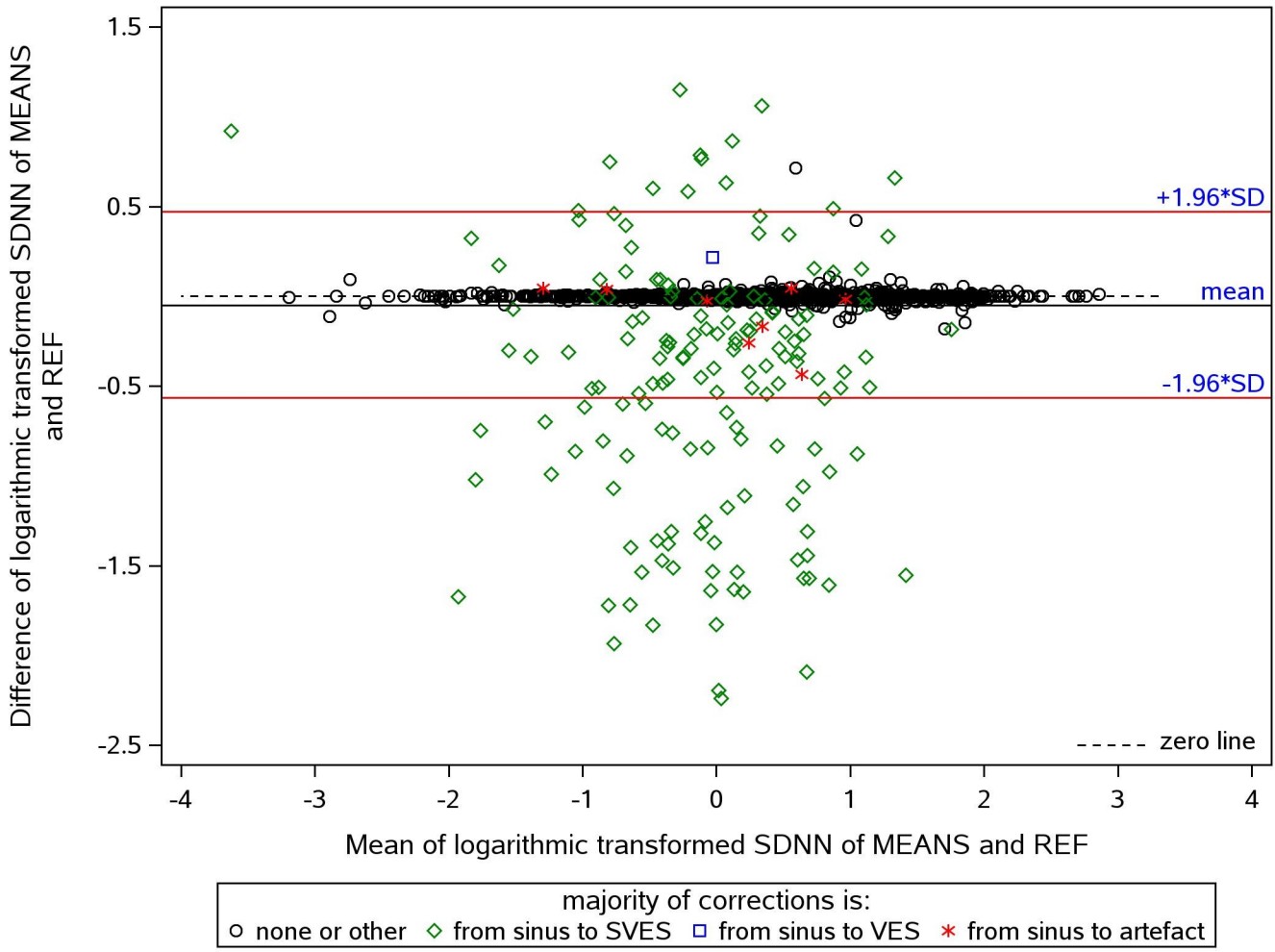

**Fig 3. Bland-Altman plot of the differences (Y-axis) and means (X-axis) of the logarithmic transformed values of LF/HF by MEANS-processed ECG versus reference standard.** Colors correspond to the majority of beat type corrections by REF in this 5-min segment.

factor. For example, the beta for the association of SDNN and sex means that men have a higher SDNN on average (0.06 on the logarithmized scale). The smaller the difference between the two beta estimators, the more independent the association is of ECG processing method.

We found that the association between HRV parameters and CVD risk factors remained essentially unaltered by the editing method both in extent and even in direction, measured as covered by CI.

The intercept in the regression of association between HRV parameters and all risk factors (except age) was greater than zero, which indicates an overestimation of HRV values by the influence of false sinus beats (values not shown in Table 5).

## 4 Discussion

Our study aimed to investigate if for HRV analyses in large population-based studies, visual editing of ECGs for ectopic beats and artifacts can be replaced by an automatic beat classification algorithm, in our case MEANS.

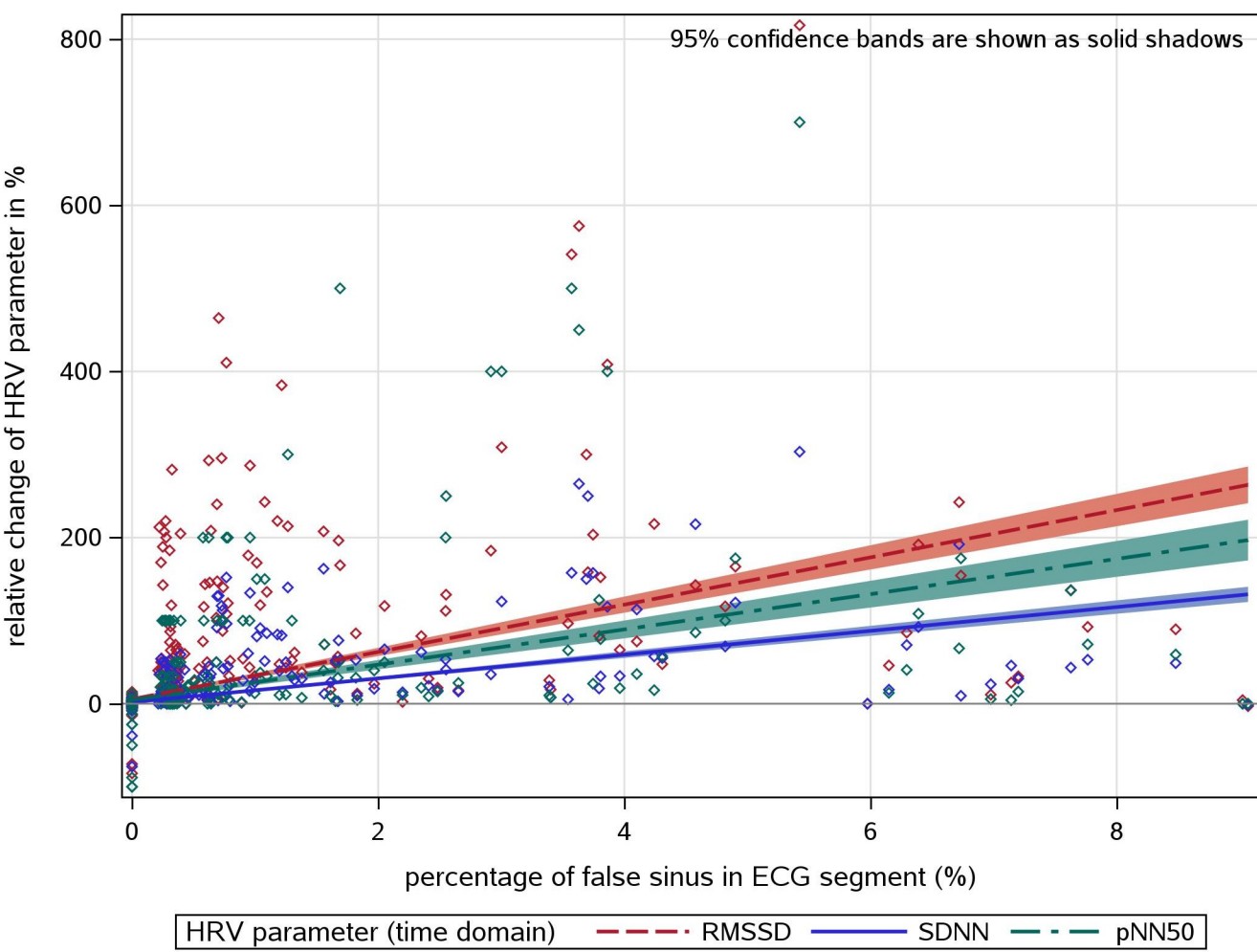

**Fig 4. Association between the proportion of false sinus beats and the relative HRV difference between MEANS only and REF for time domain HRV parameter.**

## 4.1 Performance of MEANS in discriminating sinus from non-sinus beats

We found that compared to the reference standard, MEANS classified 99% of all beats correctly. However, the average sensitivity of MEANS across all ECGs to detect SVES was low.

In their survey, Luz et al. [7] list results of sensitivity for SVES detection from different studies between 60 and 91%. In our study, we found a global sensitivity (calculated without considering individual ECGs) for SVES detection of 38%. It is therefore obvious that MEANS has some deficiency in detecting SVES (and artifacts) in single ECGs of our cohort. Whereas events classified as artifacts were mostly in reality sinus beats (in this case, the HRV values usually are not dramatically altered), the SVES were often recognized as sinus beats by MEANS (and could thus influence HRV values). Such misclassifications may have been caused by difficulties of MEANS in identifying deformed or low-amplitude P-waves. The recognition of SVES is one of the most compelling challenges for the developers of automatic algorithms at all [33].

As outlined by AAMI [8], we compared not only the global (i.e. gross statistics) but also the average performance data. In our study, average performance is noticeably higher than the global, but there is a lack of published data to compare our results with those in the literature.

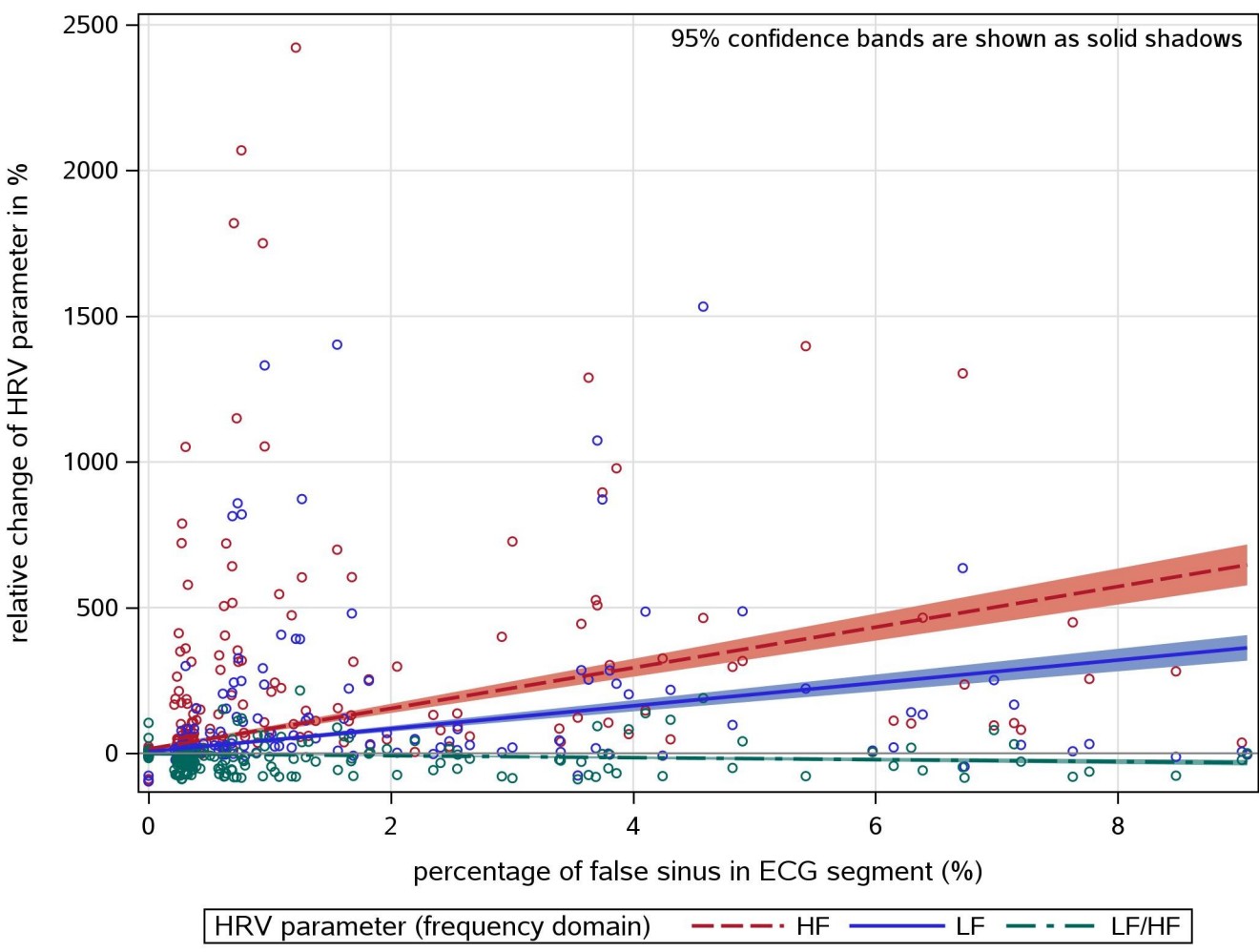

**Fig 5. Association between the proportion of false sinus beats and the relative HRV difference between MEANS only and REF for frequency domain HRV parameter.**

The difference between global and average statistics in our data is likely caused by the fact that the majority of false detections occurred in only a few ECGs. This means that in a study with participants from the general population, average performance may be more informative than

**Table 4. Association between the proportion of false sinus beats and the relative HRV difference between MEANS only and REF ($R^2$ and β estimates (per % of false sinus beats among all beats in the segment) from linear regression model).**

|  | N | $R^2$ | β estimate* | 95% CI |
|---|---|---|---|---|
| *HRV parameter* |  |  |  |  |
| SDNN | 1,640 | 0.32 | 0.14 | 0.13;0.15 |
| pNN50 | 925 | 0.20 | 0.21 | 0.18;0.24 |
| RMSSD | 1,640 | 0.24 | 0.28 | 0.26;0.31 |
| LF | 1,640 | 0.13 | 0.39 | 0.34;0.44 |
| HF | 1,640 | 0.16 | 0.70 | 0.62;0.78 |
| LF/HF | 1,640 | 0.02 | -0.03 | -0.04;-0.02 |

*- per one 1% of false sinus beats among all beats

**Table 5. Linear regression for logarithmized HRV parameters on selected risk factors and method comparison (non-parametric bootstrapping with 10.000 samples and fixed seed, 95% CI derived from percentile approach).**

| regression on risk factors and: | sex (male = reference) | | diabetes mellitus (non-diabetics = reference) | | myocardial infarction (no MI = reference) | | age (linear term) | | age (quadratic term) | |
|---|---|---|---|---|---|---|---|---|---|---|
| | beta | 95% CI | beta | 95% CI | beta | 95% CI | beta | 95% CI | beta | 95% CI |
| **ln(SDNN)** | | | | | | | | | | |
| MEANS | 0.07 | 0.02;0.12 | -0.22 | -0.29;-0.16 | -0.06 | -0.17;0.05 | -0.08 | -0.11;-0.04 | 0.0005 | 0.0003;0.0008 |
| REF | 0.06 | 0.02;0.11 | -0.24 | -0.30;-0.17 | -0.04 | -0.14;0.06 | -0.07 | -0.10;-0.04 | 0.0004 | 0.0002;0.0007 |
| bootstrapped difference | 0.01 | -0.01;0.02 | 0.01 | -0.01;0.03 | -0.02 | -0.07;0.02 | -0.01 | -0.02;0.00 | 0.0001 | 0.0000;0.0001 |
| **ln(HF)** | | | | | | | | | | |
| MEANS | 0.43 | 0.32;0.54 | -0.37 | -0.53;-0.22 | -0.20 | -0.45;0.05 | -0.25 | -0.33;-0.18 | 0.0003 | 0.0013;0.0024 |
| REF | 0.43 | 0.33;0.54 | -0.42 | -0.57;-0.28 | -0.15 | -0.39;0.08 | -0.23 | -0.30;-0.16 | 0.0016 | 0.0011;0.0022 |
| bootstrapped difference | 0.00 | -0.04;0.04 | 0.05 | -0.04;0.04 | -0.05 | -0.16;0.12 | -0.02 | -0.05;0.00 | 0.0002 | 0.0000;0.0004 |
| **ln(LF)** | | | | | | | | | | |
| MEANS | 0.01 | -0.09;0.12 | -0.53 | -0.69;-0.38 | -0.42 | -0.66;-0.18 | -0.18 | -0.25;-0.11 | 0.0011 | 0.0006;0.0017 |
| REF | 0.00 | -0.10;0.11 | -0.54 | -0.69;-0.40 | -0.39 | -0.63;-0.16 | -0.17 | -0.24;-0.10 | 0.0010 | 0.0005;0.0016 |
| bootstrapped difference | 0.01 | -0.01;0.04 | 0.01 | -0.03;0.05 | -0.02 | -0.12;0.06 | -0.01 | -0.03;0.01 | 0.0001 | 0.0000;0.0002 |
| **ln(RMMSD)** | | | | | | | | | | |
| MEANS | 0.14 | 0.08;0.20 | -0.18 | -0.26;-0.09 | 0.04 | -0.10;0.18 | -0.12 | -0.16;-0.07 | 0.0009 | 0.0006;0.0012 |
| REF | 0.14 | 0.08;0.19 | -0.21 | -0.29;-0.13 | 0.06 | -0.06;0.19 | -0.10 | -0.13;-0.06 | 0.0007 | 0.0004;0.0010 |
| bootstrapped difference | 0.00 | -0.02;0.03 | 0.03 | -0.01;0.07 | -0.02 | -0.09;0.04 | -0.02 | -0.03;0.00 | 0.0002 | 0.0000;0.0003 |
| **ln(pNN50+0.1)** | | | | | | | | | | |
| MEANS | 0.63 | -0.25;1.51 | -1.36 | -2.61;-0.11 | 1.80 | -0.16;3.76 | -1.13 | -1.73;-0.53 | 0.0086 | 0.0039;0.0132 |
| REF | 0.71 | -0.13;1.55 | -1.47 | -2.66;-0.27 | 1.87 | -0.01;3.75 | -1.04 | -1.61;-0.47 | 0.0078 | 0.0034;0.0122 |
| bootstrapped difference | -0.08 | -0.22;0.04 | 0.11 | -0.08;0.33 | -0.07 | -0.29;0.19 | -0.09 | -0.19;0.01 | 0.0008 | 0.0000;0.0016 |
| **ln(LF/HF)** | | | | | | | | | | |
| MEANS | -0.42 | -0.51;-0.34 | -0.16 | -0.28;-0.04 | -0.22 | -0.41;-0.02 | 0.08 | 0.02;0.13 | -0.0007 | -0.0012;-0.0002 |
| REF | -0.43 | -0.52;-0.35 | -0.12 | -0.24;0.00 | -0.24 | -0.43;-0.05 | 0.06 | 0.00;0.12 | -0.0006 | -0.0010;-0.0001 |
| bootstrapped difference | 0.01 | -0.01;0.04 | -0.04 | -0.09;0.00 | 0.02 | -0.01;0.05 | 0.01 | 0.00;0.03 | -0.0001 | -0.0003;0.0000 |

global performance. For SVES, we observed average sensitivity (calculated first within each ECG and then averaged) of 61%. Huang et al. [33] reported an average sensitivity of 62,8%, and Chazal et al. [34] 25,7%, which can be interpreted as a result of differences in performance, but may also be caused by differences in the study population.

## 4.2 Impact of QRS misclassifications on HRV

The fact that misclassifications of beats in the ECG cause distorted HRV values is quite common [9, 10, 35]. However, the strength of the impact was little known until recently. Few recent studies are showing that time domain parameters are more robust with respect to ectopy and artifacts than frequency domain parameters [36–39]. In contrast, Bourdillon et al. found that RMSSD and SDNN are much more sensitive to a single artifact than LF and HF [40]. Our results support the finding, that time domain parameters are more robust. Reasons for different findings, among others, could be differences in the study population and its characteristics (e.g., numerical size, age, health status) and ECG recording conditions (e.g., supine or standing, breathing conditions, short- or long-term).

Our population is from an urban elderly general population and is quite large in number by comparison. For HRV, we examine 5-min segments, i.e., short-term. We do not look at the number of ectopy or artifacts itself in the ECG but at the amount (in % related to all beats of

the segment) of beats falsely classified as sinus by our algorithm MEANS. Ectopic beats or artifacts are distributed quite unevenly across all ECGs. We consider each beat separately and do not take into account whether, for example, there is a burst of artifacts, as it is typical for motion artifacts.

Although the conclusion that in general frequency parameters are less robust to false sinus beats than time domain parameters, in our and other studies, the quotient of LF and HF seems less vulnerable to false sinus beats. However, despite that LF/HF is sometimes considered to be a measure of sympathovagal balance [41], current literature doubts this concept [25, 42–44]. It is still unclear to which physiological and psychological effects of the sympathovagal system LF/HF corresponds in detail. Furthermore, it seems to be problematic from a mathematical point of view to use a quotient of two measured quantities, which are separately very susceptible to false sinus beats.

Similar to Stapelberg et al. [38], we also observed a large spread in the relative change of HRV values. Only one or a few misclassified sinus beats can cause large changes in HRV values. In contrast, many false sinus beats may have a small effect on HRV. This could be the case if artifacts or SVES hardly disturb the sinus rhythm and thus have only a minor impact on the tachogram. On the other hand, there are examples where a single false sinus beat has strong effects, especially on RMSSD and HF.

### 4.3 Impact of QRS misclassifications on the association between HRV and CVD risk factors

We also investigated the influence of the beat classification method on the association of CVD risk factors with HRV. To our knowledge, this has never been done before. We found that using MEANS without an additional visual control does not substantially change the association between HRV (here SDNN, RMSSD, LF, HF, and quotient of LF and HF) and common CVD risk factors.

Assuming a linear relationship, it is easy to get a slope estimator for the single association. For the difference of these two estimators for both methods, we had no distributional information. Therefore, the non-parametric bootstrap and percentile method were applied [31]. To successfully implement these methods, some requirements have to be fulfilled. The most important assumption of the bootstrap procedure is that the sample must represent the population well. Since the bootstrap procedure uses only the information from the sample, the data must have enough variation in each variable to adequately represent the population. The percentile approach, which is used for computing CI, requires a not heavily skewed and approximately symmetrical probability distribution of the variable. The CARLA cohort fulfills these requirements because it is a large representative sample that reflects the characteristics of the underlying population with little selection bias, and the (logarithmized) HRV indices appear to be approximately symmetrically distributed.

### 4.4 Limitations

Our study has some limitations. Our reference was not based on the highest possible level of expertise. Instead, two trained students generated the reference standard (every ECG was assessed by one student only, without a cross-control) while being supervised by an experienced cardiologist. However, the agreement of the ratings of both students with MEANS was very similar, so that we can exclude substantial interrater differences. Furthermore, since our reference standard was a combination of a computer algorithm and visual reading, we expect even a higher validity of this reference than for an exclusively visual reading [45].

Furthermore, the within- and between-subject reliability of HRV estimates can be poor as it is affected by a range of factors during ECG recording [46]. However, in the present study, the conditions during the ECG recordings were highly controlled and standardized; thus, the measurement protocol is likely to have contributed to the improved reliability of the data. HRV parameters derived from ECGs with different durations and recording conditions are hardly comparable. Hence, we cannot extend our conclusions to setups with for example 24-h-ECGs. Additionally, we concentrated on the most crucial time- and frequency-domain parameters and did not consider others, like non-parametric parameters. Lastly, we have evaluated only one beat typification algorithm, and, therefore, our results in detail cannot be applied to every ECG processing program.

## 5 Conclusion

The automated ECG beat classification algorithm in the MEANS program appears to be suitable for ECG processing prior to HRV analysis in large population-based studies since the amount of beat misclassifications is low and its influence on the association of HRV with cardiovascular risk factors is marginal. However, it remains a challenge to improve automatic algorithms in detecting SVES or at least in identifying ECGs for additional visual reading. Machine learning and artificial intelligence systems are playing an increasingly important role in the processing of large datasets and can therefore further improve the performance of ECG analysis systems in the near future.

## Acknowledgments

We thank the work of the medical and clerical staff at the Martin-Luther-University Halle Wittenberg and all the study participants who made this study possible.

## Author Contributions

**Conceptualization:** Johannes Haerting, Karin H. Greiser, Alexander Kluttig.

**Formal analysis:** Frank Sauerbier.

**Funding acquisition:** Johannes Haerting, Karl Werdan, Karin H. Greiser.

**Methodology:** Karl Werdan, Alexander Kluttig.

**Software:** Cees A. Swenne, Jan A. Kors.

**Writing – original draft:** Frank Sauerbier.

**Writing – review & editing:** Johannes Haerting, Daniel Sedding, Rafael Mikolajczyk, Karl Werdan, Sebastian Nuding, Karin H. Greiser, Cees A. Swenne, Jan A. Kors, Alexander Kluttig.

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
