## [Decision Letter · Decision Letter 0]

21 Feb 2024

PONE-D-23-37850Impact of QRS misclassifications on heart-rate-variability parameters (results from the CARLA cohort study)PLOS ONE

Dear Dr. Sauerbier,

Thank you for submitting your manuscript to PLOS ONE. After careful consideration, we feel that it has merit but does not fully meet PLOS ONE’s publication criteria as it currently stands. Therefore, we invite you to submit a revised version of the manuscript that addresses the points raised during the review process.

We look forward to receiving your revised manuscript.

Kind regards,

Agnese Sbrollini

Academic Editor

PLOS ONE

Journal Requirements:

The CARLA Study was funded by a grant from the Deutsche Forschungsgemeinschaft (DFG) as part of the Collaborative Research Centre 598 ‘Heart failure in the elderly – cellular mechanisms and therapy’, by an additional single funding grant from the DFG, by three grants of the Wilhelm-Roux-Program of the Martin-Luther-University of Halle Wittenberg (FKZ 14/41, 16/19 and 28/21), by the Federal Employment Office, by the Ministry of Education and Cultural Affairs of Saxony-Anhalt (MK-CARLA-MLU-2011). The project “Biosalsa” (project ID: ZS/2019/07/99752) belongs to the Research Association „Autonomy in old Age“ (AiA) funded by the European Union (ERDF European Regional Development Fund) and the State of Saxony-Anhalt, Germany.

Reviewers' comments:

Reviewer's Responses to Questions

**Comments to the Author**

1. Is the manuscript technically sound, and do the data support the conclusions?

Reviewer #1: Yes

Reviewer #2: Yes

2. Has the statistical analysis been performed appropriately and rigorously? 

Reviewer #1: Yes

Reviewer #2: Yes

3. Have the authors made all data underlying the findings in their manuscript fully available?

Reviewer #1: No

Reviewer #2: Yes

4. Is the manuscript presented in an intelligible fashion and written in standard English?

Reviewer #1: Yes

Reviewer #2: Yes

5. Review Comments to the Author

Reviewer #1: The work proposes the application of the automated MEANS algorithm to detect ECG beats in a wide population of subjects. MEANS performance was compared to traditionally classified heartbeats and heart rate variability (HRV) markers were derived in both cases and compared. Findings report an overestimation of HRV when MEANS was used although this issue did not affect the association of HRV markers with risk factors in the general population analyzed.

The paper is interesting and well written, however some parts could be better defined to improve the clarity and significance of the work.

1. Abstract should better define the scope of the work and the innovation. The objectives should be reformulated in order to match the conclusions.

2. MEANS acronym should be defined.

3. In the introduction, please better describe the importance of using MEANS algorithm, in which case it was already tested and why authors decided to apply this algorithm with respect to others.

The innovation given by this work should also better defined.

5. Line 57: why “partly controversial?” please, explain.

6. Methods: it is not well clear how the random sample could be composed by 5000 subjects and the population was composed by 1779 subjects.

7. A deeper reference to the CARLA study would help the reader to focus, as well as at least a basic description of the MEANS algorithm should be added.

8. Please better define how was calculated the HRV power spectrum (e.g. with parametric or nonparametric techniques) and whether the stationarity of the series was checked before assessing the analysis and how.

9. Line 154-155: “the "gross statistics" where each beat is given equal weight (regardless of which ECG the beats are),…” the sentence is not clear, please try to rewrite.

10. In table 1 it is not clear which data are expressed as percentage or numbers.

Was there any difference between men and women?

11. Please be aware that the corresponding author listed in the journal form is different from the one listed in the pdf and must be unified.

Reviewer #2: I have reviewed the manuscript entitled 'Impact of QRS misclassifications on heart-rate-variability parameters(results from the CARLA cohort study)'.

The role or HRV parameters for the prediction of sympatethic and autonomic tone should be mentioned in the intro section citing its role in post covid era. Please consider citing 'Heart rate variability and cardiac autonomic functions in post-COVID period'. Moreover HRV paramateres can significantly change with healthylifestyle therefore the interpretation of these parameters plays an important role in directing patients. This should be briefly mentioned in the intro part citing 'Effect of a mobile application and smart devices on heart rate variability in diabetic patients with high cardiovascular risk: A sub-study of the LIGHT randomized clinical trial'.

Machine learning and artificial intelligence systems play very important role in order to reach precise results while interpreting the misclassifications of QRS complexes. Please consider citing 'The Role of Artificial Intelligence in Coronary Artery Disease and Atrial Fibrillation' in the discussion section while mentioning the role of AI

6. PLOS authors have the option to publish the peer review history of their article (what does this mean?). If published, this will include your full peer review and any attached files.

Reviewer #1: No

Reviewer #2: No

---

## [Author Response · Author response to Decision Letter 0]

5 Apr 2024

Answer to Editor's questions:

Author response: We tried to fulfill all requirements of the formatting guidelines.

Author response: We have revised and expanded the "Funding Information" section in editorialmanger.

The CARLA Study was funded by a grant from the Deutsche Forschungsgemeinschaft (DFG) as part of the Collaborative Research Centre 598 ‘Heart failure in the elderly – cellular mechanisms and therapy’, by an additional single funding grant from the DFG, by three grants of the Wilhelm-Roux-Program of the Martin-Luther-University of Halle Wittenberg (FKZ 14/41, 16/19 and 28/21), by the Federal Employment Office, by the Ministry of Education and Cultural Affairs of Saxony-Anhalt (MK-CARLA-MLU-2011). The project “Biosalsa” (project ID: ZS/2019/07/99752) belongs to the Research Association „Autonomy in old Age“ (AiA) funded by the European Union (ERDF European Regional Development Fund) and the State of Saxony-Anhalt, Germany.

Author response: Thanks for clarification. The cover letter (please see above) has been supplemented by the relevant statement.

Answers to the reviewer's questios:

5. Review Comments to the Author

Reviewer #1: The work proposes the application of the automated MEANS algorithm to de-tect ECG beats in a wide population of subjects. MEANS performance was compared to tra-ditionally classified heartbeats and heart rate variability (HRV) markers were derived in both cases and compared. Findings report an overestimation of HRV when MEANS was used alt-hough this issue did not affect the association of HRV markers with risk factors in the general population analyzed.

The paper is interesting and well written, however some parts could be better defined to improve the clarity and significance of the work.

Author response: Thank you!

1. Abstract should better define the scope of the work and the innovation. The objectives should be reformulated in order to match the conclusions.

Author response: Thanks for the suggestion. We have revised the abstract to better align tasks and objectives, unfortunately the number of characters available in the abstract is lim-ited. It now reads (line 22):

The Modular ECG Analysis System MEANS is a widely used tool, especially in large studies. The aim of this study was therefore to evaluate MEANS for its ability to detect non-sinus ECG beats and arti-facts and to compare HRV parameters in relation to ECG processing.

Additionally, we analyzed how ECG processing affects the statistical association of HRV with cardio-vascular disease (CVD) risk factors.

2. MEANS acronym should be defined.

Author response: MEANS as acronym is now already defined in the abstract (line 22).

3. In the introduction, please better describe the importance of using MEANS algorithm, in which case it was already tested and why authors decided to apply this algorithm with re-spect to others.

The innovation given by this work should also better defined.

Author response: Thank you for pointing this out. We extended the last section in the intro-duction starting with line 70:

„Therefore, our aim was i) to evaluate the performance of a widely used algorithm in the classification of heart-beats in a general population, ii) to quantify the effect of undetected non-sinus ECG beats on the calculation of HRV parameters, and iii) to analyze how these misclassifications affect the statistical association of HRV with CVD risk factors.

The "Modular ECG Analysis System" (MEANS) is a validated (11) and widely used automatic algorithm in popula-tion-based studies, e.g., the Rotterdam study (12), the HELIUS study (13), the SHIP study (14), and the NAKO study (15). The development of the program began several decades ago. It has a completely modular structure. Individ-ual modules could therefore be entirely redeveloped without disrupting the overall framework. In a testing of nine different programs, MEANS showed excellent results (16). However, the dependence of the calculated HRV on such an ECG analysis system and, in particular, the dependence of association studies involving HRV, have not yet been investigated. We, therefore, decided to perform the analysis using the MEANS algorithm as an important example for heartbeat classification algorithms. “

5. Line 57: why “partly controversial?” please, explain.

Author response: We have deleted this sentence as the statement is of little relevance to our investigation.

6. Methods: it is not well clear how the random sample could be composed by 5000 subjects and the population was composed by 1779 subjects.

Author response: Thanks for the suggestion. The first paragraph of chapter two has been significantly revised and expanded to make the recruitment process clearer. Starting with line 90 it now reads:

“A random sample of 5.000 people aged 45 to 80 years at the time of the sampling (July 2002) was drawn from the population registry of the city of Halle. The recruitment of study subjects has been done by inviting consecutive waves of random sub-samples of the original population sample. Accordingly, not all persons originally drawn from the population registry had to be invited in order to achieve a representative target sample of 1750 subjects of the Halle population aged 45–80 years. Of the 3437 subjects invited to participate in the study, 1779 partici-pants aged 45-83 years at baseline were recruited, of which 812 (46%) were women and 967 (54%) men, resulting in a final response proportion of 64.1% after exclusion of persons who deceased prior to the invitation, moved away or were unable to participate due to illness. The recruitment of study subjects and the baseline examination began in December 2002 and ended in January 2006. All participants gave their written informed consent.”

7. A deeper reference to the CARLA study would help the reader to focus, as well as at least a basic description of the MEANS algorithm should be added.

Author response: Thanks for the suggestion. As with the last point we revised and expanded accordingly. Starting with line 84 it now reads:

“The analyses are based on data from the baseline examination of the prospective, population-based CARLA study (CARdiovascular disease, Living and Ageing in Halle). The original aim of the CARLA study was to analyze the causes of increased cardiovascular morbidity and mortality in the region and to investigate the importance of HRV as a marker of autonomic function and predictor of cardiovascular events. Details of the study have been described elsewhere (17-19). In brief, the CARLA study is a prospective cohort study of a representative sample of the elderly inhabitants of the city of Halle (Saale).”

And line 117:

As already mentioned, MEANS has a completely modular structure. Signal evaluation is mainly carried out serially by the modules. In addition to the input and output modules, 4 further module groups are responsible for the following tasks [11]:

• detection of QRS complexes and disturbances as well as the P waves

• typification according to waveform for the QRS or ST-T complexes

• determination of the dominant complexes for contour analysis

• segmentation of P and T waves and QRS complexes (recognition of start and end points)

• classification (rhythm, contour) of the ECG.

8. Please better define how was calculated the HRV power spectrum (e.g. with parametric or nonparametric techniques) and whether the stationarity of the series was checked before assessing the analysis and how.

Author response: Thanks for the hint. We extended the part about testing stationarity. Fre-quency parameters were calculated using (old-fashioned) FFT, which is a non-parametric technique. The extended part starts at line 140:

Additionally, the remaining segments were checked for stationarity of the RR intervals according to the reverse arrangement test at the 5% level to means and variances and removed when the stationarity z-score cut-off ex-ceeded 1.96 (24). Removal of the segments with non-stationary data did not lead to the exclusion of any of the study participants due to a lack of data. From the remaining segments, the segment of the MEANS-processed ECG with the lowest percentage of non-sinus beats and the corresponding REF segment was selected for HRV analysis. If several choices were possible, the earliest segment occurring in time was chosen.

9. Line 154-155: “the "gross statistics" where each beat is given equal weight (regardless of which ECG the beats are),…” the sentence is not clear, please try to rewrite.

Author response: Thanks for the suggestion. We tried to make the sense clearer by re-writing. Starting with line 174:

“Performance analyses of MEANS regarding the correct detection of non-sinus beats were done using the entire 20-min ECGs. The sensitivity and specificity of the non-sinus beat detection performance of MEANS were deter-mined according to the AAMI guidelines (8). Two types of aggregated statistics are given: in the "gross statistics", all beats are weighted equally, regardless of whether they originate from one or the other ECG. The dependency of beats from one ECG is therefore not taken into account. This is in contrast to the average statistics, in which the dependency of beats from an ECG is taken into account and beat statistics are calculated individually for each ECG and then averaged.“

10. In table 1 it is not clear which data are expressed as percentage or numbers.

Was there any difference between men and women?

Author response: Thanks for the hint. We have reformatted the table to make the units clearer. The table shows substantial differences between men and women with regard to body weight and prevalence of myocardial infarction. However, these differences do not play a role in our study, which is why we have not addressed them in the manuscript.

11. Please be aware that the corresponding author listed in the journal form is different from the one listed in the pdf and must be unified.

Author response: Thank you for bringing this to our attention. We changed the correspond-ing author in the editorial manager system accordingly. 

Reviewer #2: I have reviewed the manuscript entitled 'Impact of QRS misclassifications on heart-rate-variability parameters (results from the CARLA cohort study)'.

The role or HRV parameters for the prediction of sympatethic and autonomic tone should be mentioned in the intro section citing its role in post covid era. Please consider citing 'Heart rate variability and cardiac autonomic functions in post-COVID period'.

Author response: Thank you for this hint. We have expanded the sentence on the im-portance of HRV accordingly and cited your publication (starting line 55).

Moreover HRV paramateres can significantly change with healthylifestyle therefore the in-terpretation of these parameters plays an important role in directing patients. This should be briefly mentioned in the intro part citing 'Effect of a mobile application and smart devices on heart rate variability in diabetic patients with high cardiovascular risk: A sub-study of the LIGHT randomized clinical trial'.

Author response: Thank you for pointing this out. We have expanded the sentence on the importance of HRV accordingly.

Machine learning and artificial intelligence systems play very important role in order to reach precise results while interpreting the misclassifications of QRS complexes. Please con-sider citing 'The Role of Artificial Intelligence in Coronary Artery Disease and Atrial Fibrilla-tion' in the discussion section while mentioning the role of AI

Author response: Thank you for this hint. We have extended our conclusion (Conclusion, last sentences). To be honest, two of the publications mentioned seem to us to have a little or no relevance to the topic of our study. We have not taken all of them into account.

---

## [Decision Letter · Decision Letter 1]

21 May 2024

Impact of QRS misclassifications on heart-rate-variability parameters (results from the CARLA cohort study)

PONE-D-23-37850R1

Dear Dr. Kluttig,

We’re pleased to inform you that your manuscript has been judged scientifically suitable for publication and will be formally accepted for publication once it meets all outstanding technical requirements.

Kind regards,

Agnese Sbrollini

Academic Editor

PLOS ONE

Reviewer's Responses to Questions

**Comments to the Author**

1. If the authors have adequately addressed your comments raised in a previous round of review and you feel that this manuscript is now acceptable for publication, you may indicate that here to bypass the “Comments to the Author” section, enter your conflict of interest statement in the “Confidential to Editor” section, and submit your "Accept" recommendation.

Reviewer #1: All comments have been addressed

Reviewer #2: All comments have been addressed

2. Is the manuscript technically sound, and do the data support the conclusions?

Reviewer #1: Yes

Reviewer #2: Yes

3. Has the statistical analysis been performed appropriately and rigorously? 

Reviewer #1: Yes

Reviewer #2: Yes

4. Have the authors made all data underlying the findings in their manuscript fully available?

Reviewer #1: No

Reviewer #2: No

5. Is the manuscript presented in an intelligible fashion and written in standard English?

Reviewer #1: Yes

Reviewer #2: Yes

6. Review Comments to the Author

Reviewer #1: After the revision, authors have addressed the reviewer's comments and the work havs been improved in clarity and significance. I have no further request.

Reviewer #2: I have reviewed the manuscript entitled 'Impact of QRS misclassifications on heart-rate-variability parameters

(results from the CARLA cohort study)'.

The manuscript is acceptable after the revisions.

7. PLOS authors have the option to publish the peer review history of their article (what does this mean?). If published, this will include your full peer review and any attached files.

Reviewer #1: No

Reviewer #2: No
